# Salivary miRNA Profiles in COVID-19 Patients with Different Disease Severities

**DOI:** 10.3390/ijms241310992

**Published:** 2023-07-01

**Authors:** Irma Saulle, Micaela Garziano, Gioia Cappelletti, Fiona Limanaqi, Sergio Strizzi, Claudia Vanetti, Sergio Lo Caputo, Mariacristina Poliseno, Teresa Antonia Santantonio, Mario Clerici, Mara Biasin

**Affiliations:** 1Department of Pathophysiology and Transplantation, University of Milan, Via Francesco Sforza, 20122 Milan, Italy; irma.saulle@unimi.it (I.S.); micaela.garziano@unimi.it (M.G.); fiona.limanaqi@unimi.it (F.L.); mario.clerici@unimi.it (M.C.); 2Department of Biomedical and Clinical Sciences, University of Milan, Via G.B. Grassi, 20122 Milan, Italy; gioia.cappelletti@unimi.it (G.C.); sergio.strizzi@unimi.it (S.S.); claudia.vanetti@unimi.it (C.V.); 3Unit of Infectious Diseases, Department of Clinical and Experimental Medicine, University of Foggia, 71122 Foggia, Italy; sergiolocaputo@gmail.com (S.L.C.); polisenomc@gmail.com (M.P.); teresa.santantonio@unifg.it (T.A.S.); 4Don C. Gnocchi Foundation, Istituto di Ricovero e Cura a Carattere Scientifico (IRCCS) Foundation, Via A. Capecelatro 66, 20148 Milan, Italy

**Keywords:** miRNA, non-coding RNA, COVID-19, SARS-CoV-2, epigenetic profile

## Abstract

The oral mucosa is the first site of SARS-CoV-2 entry and replication, and it plays a central role in the early defense against infection. Thus, the SARS-CoV-2 viral load, miRNAs, cytokines, and neutralizing activity (NA) were assessed in saliva and plasma from mild (MD) and severe (SD) COVID-19 patients. Here we showed that of the 84 miRNAs analyzed, 8 were differently expressed in the plasma and saliva of SD patients. In particular: (1) miRNAs let-7a-5p, let-7b-5p, and let-7c-5p were significantly downregulated; and (2) miR-23a and b and miR-29c, as well as three immunomodulatory miRNAs (miR-34a-5p, miR-181d-5p, and miR-146) were significantly upregulated. The production of pro-inflammatory cytokines (IL-1β, IL-2, IL-6, IL-8, IL-9, and TNFα) and chemokines (CCL2 and RANTES) increased in both the saliva and plasma of SD and MD patients. Notably, disease severity correlated with NA and immune activation. Monitoring these parameters could help predict disease outcomes and identify new markers of disease progression.

## 1. Introduction

SARS-CoV-2 is an airborne pathogen transmitted by asymptomatic, pre-symptomatic, and symptomatic individuals through close contact via exposure to infected droplets and aerosols [1,2]. Saliva plays a key role in virus spread and transmission. However, saliva is also critical in protecting the oral cavity against microorganisms, as it harbors several components that intervene in the early host defense against invasive pathogens. Thus, saliva composition and function can be critical for determining the course of the infection. Immune-associated factors present in saliva include neutralizing antibodies, chemokines, and cytokines, as well as epigenetic factors such as microRNA (miRNA), which can direct the host response through the modulation of gene expression. In line with this, recent studies corroborated the diagnostic value of salivary miRNAs as biomarkers of disease [3,4,5,6].

MiRNAs are noncoding (18–25 nt length) intracellular molecules that downregulate or repress the expression and/or translation of protein-coding genes by binding the 3-untranslated region (3′-UTR) and the coding sequence of mRNA molecules [7]. MiRNAs can also be found in the extracellular space, including biofluids such as plasma and saliva [8], where they act as signaling molecules to regulate a broad array of physiological and pathological processes. In viral infections, several studies argue that miRNAs can promote viral inactivation and neutralization; this might occur by directly repressing viral RNA translation and/or by targeting host genes to improve the immunological response against the virus [9]. As for SARS-CoV-2, ex vivo studies documented an altered plasma miRNA profile in different COVID-19 cohorts [10]. As assessed by computational prediction studies, these different profiles could mirror distinct phases of the disease, from its onset to recovery [9].

Studies assessing miRNA profiles in the oral mucosa of COVID-19 patients are very limited [11]. We verified whether SARS-CoV-2 infection affects the composition and the expression of salivary miRNA and if such an effect is different in patients with diverse degrees of COVID-19 severity. We also analyzed other salivary immune parameters in the same patients in the attempt to define salivary immune profiles that could predict COVID-19 severity. Finally, we verified whether SARS-CoV-2 infection induces similar alterations of immune profiles in saliva and plasma. Results suggest that salivary miRNA molecular phenotyping could represent a valuable, non-invasive diagnostic biomarker.

## 2. Results

### 2.1. Study Population

A total of 20 COVID-19-positive patients were enrolled in the study, 10 with mild disease and 10 with severe disease, divided according to COVID-19 disease severity following the National Institute of Health (NIH) guidelines for COVID-19 treatment. In particular:

*Mild disease (MD)* refers to individuals who had any of the various signs and symptoms of COVID-19 (e.g., fever, cough, sore throat, malaise, headache, muscle pain, nausea, vomiting, diarrhea, and loss of taste and smell) but did not have shortness of breath, dyspnea, or abnormal chest imaging.

*Severe disease* refers to individuals who had an SpO_2_ < 94% in room air at sea level, a ratio of arterial partial pressure of oxygen to fraction of inspired oxygen (PaO_2_/FiO_2_) < 300 mm Hg, a respiratory rate > 30 breaths/min, or lung infiltrates > 50%.

Hydroxychloroquine was prescribed to 47 patients (80%), 22 of whom were treated with a protease inhibitor (Lopinavir/RTV, 9 pts, or Darunavir/COBI, 10 pts), in line with the standard of care at the time of the patients’ hospitalization. Five patients (4%) also received Tocilizumab in combination with one of the aforementioned treatments. Thirteen patients (16%) did not receive any therapy.

Oxygen was administered to 44 patients (75%), the majority of whom (27 pts, 61%) received oxygen therapy in a Venturi mask. Six patients (14%) required high-flow nasal (HFN) oxygen support, and five (11%) required non-invasive ventilation (NIV). Seven patients (12%) with severe SARS-CoV-2 pneumonia were transferred to the ICU; three of them received mechanical ventilation. We chose ten of thirteen patients who did not receive any therapy. We excluded 3 of those 13 patients because they lacked some biological samples. As for MD patients, non-smokers and patients without comorbidities were considered for analysis. The 10 HC were non-smoker subjects. All patients and HC were matched for age and sex (Table 1).

### 2.2. SARS-CoV-2 Titers and Neutralizing Activity (NA) in Saliva and Plasma

To verify if COVID-19 clinical severity was correlated with viral replication, we evaluated SARS-CoV-2 viral titers in all saliva and plasma samples. Not unexpectedly, higher salivary viral titers were detected in SD patients, compared to MD patients, for both N1 and N2 (*p* < 0.05 for both comparisons) (Figure 1A), with SARS-CoV-2 being totally undetectable in the saliva of HC.

To verify whether the different viral replication rates seen in SD patients, compared to MD patients, could be associated with different salivary and/or plasma neutralization activities, a neutralization assay was performed next on all saliva. Results showed that neutralization activity was indeed more potent in saliva samples from MD patients, compared to SD patients (*p* < 0.05) (Figure 1B), suggesting that a stronger NA at the mucosal level is able to properly contain viral replication and spread, thus avoiding the onset of severe symptoms.

Notably, a comparison of the strength of NA in saliva with viral titers in all the subjects enrolled in the study showed the presence of a statistically significant negative correlation between these two parameters (*p* < 0.05) (Figure 1D).

In contrast with these results, no differences were observed when plasma NAs from MD and SD patients were compared (Figure 1C).

### 2.3. MiRNA Expression in Saliva from SD Patients, MD Patients, and HC

The expression of 84 miRNAs known to be endowed with immunomodulatory and/or antiviral functions was assessed by real-time qPCR in the saliva of all the enrolled subjects. Results showed that SD patients were characterized by a peculiar miRNA profile that was different, compared to that of MD patients and HC (Figure 2A) (Appendix A).

In an attempt to better define the miRNA profile associated with severe SARS-CoV-2 infection, we focused on antiviral and immunomodulatory miRNAs, whose expression was differentially modulated in SD patients, compared to MD patients and HC (Figure 2B,C). In SD patients, compared to MD patients and HC, several members of the let-7 antiviral family, including let-7a, let-7b, and let-7c, were downregulated (*p* < 0.05 for all comparisons), whereas miR-23 and miR-29c were significantly upregulated (miR-23: *p* < 0.05; miR-29c: *p* < 0.01, Figure 2B). As for miRNAs known to be endowed with an immunomodulatory function, miR-34, miR-146, and miR-181 were significantly upregulated in SD patients, compared to both MD patients and HC (miR-34: *p* < 0.01, miR-146: *p* < 0.05, miR-181: *p* < 0.05) (Figure 2C). Notably, miR-34 and miR-146 were overexpressed in MD patients, compared to HC (miR-34: *p* < 0.05 and miR-146: *p* < 0.05) as well, suggesting that the modulation of miRNA salivary profiles occurs even at low viral replication rates.

### 2.4. MiRNA Expression in Plasma from SD Patients, MD Patients, and HC

The same miRNA panel was evaluated in plasma collected from all the individuals enrolled in the study. The expressions of the 84 miRNAs analyzed in plasma were different in the three groups examined, with a significant up-regulation of a number of miRNAs characterizing SD patients (Figure 3A) (Appendix A). The miRNA row data are available on NCBI platform with GSE accession number: GSE236017.

In SD patients in particular, both antiviral (vs. MD: miR-23a: *p* < 0.05; miR-23c: *p* < 0.01, miR-29c: *p* < 0.05, and miR-98: *p* < 0.05) and immunomodulatory (vs. MD: miR-34a, miR-146, and miR-181: *p* < 0.05 for all comparisons) (vs. HC: miR-34a, miR-146, and miR-181: *p* < 0.05 for all comparisons) miRNAs were significantly upregulated (Figure 3B,C). In addition, miR-23a, miR-29c, and miR-181 (*p* < 0.05 in all cases) were upregulated in MD patients, compared to HC. Finally, significant downregulations of let-7 family members were observed in SD patients (let-7a, let7b, and let-7f: *p* < 0.05 in all cases) and MD patients (let-7a: *p* < 0.05; let-7b: *p* < 0.05; let-7f: *p* < 0.01), compared with HC, and for let-7f in SD versus MD patients (*p* < 0.05) (Figure 3B).

Notably, most miRNAs were similarly modulated by infection in saliva and plasma, validating the assumption that their regulation is a virus-specific event and highlighting their potential as biomarkers to monitor viral infection.

### 2.5. Gene Expression of Immune/Antiviral-Selected Effectors in PBMCs from SD and MD Patients

We evaluated the mRNA expressions of 40 selected genes in PBMCs of SD and MD patients. In SD patients, compared to MD patients, the expression pattern was characterized by a significant increase in the expression of: (1) activation markers (CD69: *p* < 0.05; CD38: *p* < 0.01); (2) pro-inflammatory (IL-1β, IL-6, and IL-22: *p* < 0.05 for all comparisons) and anti-inflammatory cytokines (IL-10: *p* < 0.05); (3) chemokines (CCL2 and CCL5: *p* < 0.05), and (4) host antiviral effector genes (IFITM1 and IFITM3: *p* < 0.01) (Figure 4). These results offer further confirmation to the observation that an overstimulation of the immune response is seen in severe COVID-19 infection, a situation where antiviral mechanisms are exacerbated as well.

### 2.6. Modulation of Cytokine and CHEMOKINE Productions in Saliva and Plasma from SD Patients, MD Patients, and HC

The production of 27 cytokines and chemokines related to immune activation was assessed in both plasma and saliva specimens from all the subjects with a Luminex assay. The cytokine and chemokine patterns were similar in the two anatomical districts and were characterized by the upregulation of all analyzed proteins in SD patients, compared to both MD patients and HC. However, the upregulation of specific cytokines reached statistical significance only in the plasma of the patients analyzed. In fact, peculiar differences between the analyzed cohort were far more evident in plasma than in saliva, probably reflecting the systemic cytokine storm raging in patients with severe symptoms.

In detail, in the saliva of SD patients, such factors included cytokines known to be involved in the inflammatory process caused by COVID-19, such as IL-1β (SD vs. HC: *p* < 0.01; vs. MD: *p* < 0.05), IL-2 (SD vs. HC: *p* < 0.05), IL-6 (SD vs. HC and MD: *p* < 0.05), IL-8 (SD vs. HC: *p* < 0.001; vs. MD: *p* < 0.05), IL-9 (SD vs. HC: *p* < 0.05), and TNFα (SD vs. HC: *p* < 0.01). Some chemokines, including RANTES (SD vs. HC: *p* < 0.01; vs. MD: *p* < 0.05) and CCL2 (SD vs. HC: *p* < 0.01; vs. MD: *p* < 0.05), as well as growth factors such as VEGF (SD vs. HC: *p* < 0.01) (Figure 5A), were significantly increased in SD patients as well.

Notably, saliva concentrations of a number of immune proteins were also increased in MD patients, compared to HC (IL-1β: *p* < 0.01; IL-6: *p* < 0.05; IL-8: *p* < 0.01; IL-9: *p* < 0.01; CCL-2: *p* < 0.001; RANTES: *p* < 0.05; TNFα: *p* < 0.05; and VEGF: *p* < 0.01) (Figure 5A).

Likewise, in plasma, the concentrations of a number of immune proteins were significantly increased in SD patients as follows: IL-1β (vs. HC and MD: *p* < 0.05), IL-6 (vs. HC and MD: *p* < 0.01), IL-8 (vs. HC: *p* < 0.01) IL-10 (vs. HC: *p* < 0.05), IL-17 (vs. HC and MD: *p* < 0.01), IFNγ (vs. HC: *p* < 0.05), VEGF (vs. HC: *p* < 0.001), CCL-2 (vs. HC and MD: *p* < 0.01), TNFα (vs. HC and MD: *p* < 0.05), and RANTES (vs. MD: *p* < 0.05).

Finally, in plasma samples from MD patients, compared to HC, significantly increased productions of IL-1β (*p* < 0.05), IL-6 (*p* < 0.05), IL-8 (*p* < 0.05), IL-10 (*p* < 0.05), IL-17 (*p* < 0.01), IFNγ (*p* < 0.05), CCL-2 (*p* < 0.001), VEGF (*p* < 0.001), and TNFα (*p* < 0.05) were observed as well (Figure 5B).

## 3. Discussion

Circulating miRNAs are endogenous, non-coding small RNA molecules that can be secreted into circulation and biological fluids such as saliva. Similar to intercellular miRNAs, circulating ones partake in the regulation of both biological processes and the control or progression of diseases, including infections [12,13,14]. Indeed, over the course of infection, pathogens trigger a significant change in the signatures of cellular and/or circulating miRNAs, suggesting their use as circulating biomarkers of disease [12]. Due to collection simplicity and because of extended similarities between mucosal and systemic miRNA profiles, recent studies have focused on the convenience of the saliva-based liquid biopsy approach for multiple sampling, early diagnosis, prognosis, longitudinal monitoring of progression, and treatment response in different pathological conditions [15,16,17].

Only a few studies so far have focused on identifying saliva biomarkers that can predict different clinical outcomes in COVID-19 patients [18,19], and our data represent the first description of epigenetic and immunological determinants at the entry site of SARS-CoV-2 in saliva of COVID-19 patients prior to vaccination.

Overall, we observed a complex dysregulation of soluble immune factor profiles following SARS-CoV-2 infection at both systemic and local levels, with an intensity proportional to disease severity. Among all the analyzed cytokines and miRNAs, we focused on those showing a superimposable trend in saliva and plasma specimens in an attempt to identify valuable biomarkers for monitoring disease progression.

In particular, in COVID-19 patients, we detected a significant down-regulation of let-7 miRNA family members in both anatomical districts. Let-7 is a miRNA family that includes 13 members with established antiviral activity. In fact, it has been reported that let-7 can attenuate the virulence of the influenza virus, preventing the development of pneumonia [20], a well-known recurrent feature of SARS-CoV-2 infection [6]. Even more noteworthy, let-7d, let-7e, let-7f, let-7g, let-7i, and miR-98 were reported to significantly suppress SARS-CoV-2 spike protein production and/or inhibit membrane protein assembly by directly targeting SARS-CoV-2 [21]. Further endorsing these observations, lower levels of let-7 family members were observed in plasma from severe COVID-19 patients, compared to mild ones and healthy controls [22], and their expression was further reduced in patients requiring oxygen [10]. Our data in saliva specimens are, therefore, in line with the results obtained in plasma by other researchers, suggesting their potential use as biomarkers to monitor SARS-CoV-2 infection at the entry site.

Conversely, in our cohort, miR-23 and miR-29 expressions were significantly up-regulated in both plasma and saliva of SD patients. MiR-29c is one of the most important antiviral microRNAs associated with different disease outcomes, depending on the infectious agent. We observed that miR29c expression was significantly higher in PBMCs and plasma from subjects who did not seroconvert, despite repeated exposure to HIV-1, suggesting a protective role for this miRNA against HIV-1 infection. Likewise, an increased miR-29 expression was recently detected in placentas from SARS-CoV-2-infected pregnant women [23] in whom vertical SARS-CoV-2 transmissions were not observed. As this miRNA was predicted to target 11 SARS-CoV-2 sites [24], these results suggest a potential defensive role of miR-29 in preventing vertical transmission.

Contrariwise, other studies hypothesized a negative role for miR-29, as its expression was increased in PBMCs of COVID-19 patients, compared to healthy controls during the acute and post-acute phases [25] and in the plasma of severe patients, compared to mild patients [26].

As for miR-23, bioinformatics analysis predicted its binding to the 3′UTR of the SARS-CoV-2 spike protein and the ACE2 cellular receptor [27], and a high concentration of this miRNA could classify SARS-CoV-2 infection with >99% accuracy [10]. The increased expression of this molecule we detected in both SD plasma and saliva could, therefore, be explained as an attempt of the host to counteract SARS-CoV-2 infection once viral replication exceeds a definite threshold.

As expected, in our cohort, SARS-CoV-2 infection triggered the dysregulation of several immunomodulatory miRNAs as well. Indeed, as documented by previous studies, miRNAs can either enable viral immune evasion through targeting some pivotal host immune reactions [28] or decrease host responses to prevent acute tissue damage by targeting immunological mediators [29]. Notably, the upregulations of miR-34, miR-146, and miR-181 in COVID-19 patients were evident in both plasma and saliva. In post-mortem lung biopsies from COVID-19 patients, endothelial dysfunction was associated with miR-34a downregulation [30]. Instead, Chen et al. observed that in septic mice, miR-34 over-expression promoted oxidative stress, pyroptosys, and the consequent production of pro-inflammatory cytokines, eventually resulting in lung injury [31]. Our findings support the hypothesis of Chen and colleagues, which state that miR-34a overexpression is a marker of COVID-19 severity.

The role of miRNA 146 in SARS-CoV-2 infection remains largely controversial, as contradictory results have been published. It acts as a dominant negative regulator of the innate immune response [32] by decreasing NK cell degranulation [33] and the expression of downstream factors of TLR signaling [34]. Switching off the immune response is a double-edged sword with unforeseeable consequences. Hence, in some viral infections, miR-146 overexpression is associated with an increased viral replication [35]. In severe COVID-19 patients, miR-146 deregulation was linked to an over-activation of the immune system, causing the cytokine storm [36]. Conversely, and in line with our results, some authors detected a higher expression of miR-146 in severe patients [37] and in the acute and post-acute COVID-19 phases [38], thus ascribing it a negative role. This notwithstanding, in another cohort of severe COVID-19 patients, high levels of miR-146 were associated with a lower thrombotic risk and mortality, due to an inverse correlation with D-dimer formation [39].

The precise function of miR-181 in SARS-CoV-2 infection is still debated as well. miR-181-5p was shown to block immune checkpoints by targeting Bcl2 and TNF super families [40]. An increase in this miRNA was reported in patients with severe COVID-19 [41], and its expression was associated with a worse outcome [42]. In our cohort, an un-upregulation of miRNA-181 was detected in both SD saliva and plasma as well. On the contrary, recent studies reported that miR-181a levels are usually downregulated in SARS-CoV-2-infected patients, and low levels of miR-181a boost viral entry, as it targets TMPRSS2 and ACE2 expressions [43,44].

Notably, the miRNA data were not SARS-CoV-2-specific, or, at least, our data could not prove a direct correlation between SARS-CoV-2 infection and miRNA production. Most of these miRNAs display antiviral or immunological functions. Indeed, their expression has already been widely documented in other pathological conditions, including infectious ones [16,17,27,45]. We, therefore, speculate that this miRNA outline is triggered by the host response to any invading pathogen, rather than a specific one. However, as their modulation was significantly different in COVID-19 patients, compared to HC and in patients with different disease severities, it would be very useful to identify a miRNA combination suggestive of SARS-CoV-2 infection and outcome, especially in the saliva, to be used as predictive prognosis and diagnosis tools.

Because of their antiviral and immunomodulatory functions, miRNAs could also directly intervene in the modulation of NA. Although we did not directly validate this hypothesis, our data showed a correlation between disease severity and NA, which could be at least partially attributed to miRNA release, mainly at the mucosal level. Indeed, in the saliva of MD patients, we observed a robust NA, suggesting that the activation of a prompt and protective immune response prevents the onset of a severe symptomatology. Intriguingly, the NA trend observed in the saliva of MD vs. SD patients was opposite to that detected in the plasma. This inverse tendency has already been documented in other viral infections. For instance, Nekoua et al. reported that in coxsackievirus infection, NA was significantly higher in saliva from diabetic patients, compared to controls, but such differences were no longer evident at the plasma level [46].

It is tempting to speculate that the levels of NA in the plasma mirror the strong and generalized immune reaction (cytokine storm) that occurs at a systemic level following viral infection and tissue dissemination. Thus, it is conceivable that plasma NA levels are proportional to viral titers and to the severity of disease, being higher in SD patients, compared to MD patients. Instead, NA in the saliva likely mirrors the pattern of an early, local immune response focused at controlling viral spread and tissue damage. In this case, the levels of NA might be inversely proportional to the severity of disease, being higher in MD patients, compared to SD patients. However, the lack of statistical significance in salivary NA between MD and SD patients hinders any further speculations.

Accordingly, a paper from Valimaa et al. showed that in Herpes simplex virus (HSV) infection, NA in saliva from asymptomatic HSV-seropositive subjects was significantly higher, compared to saliva NA in patients with recurrent labial herpes, suggesting that salivary defense factors might contribute to controlling the recurrence and, presumably, the severity of infections [47].

In the plasma of SD patients, we also detected higher levels of pro-inflammatory cytokines/chemokines possibly responsible for the well-known cytokine storm. Notably, the same scenario was recorded by profiling salivary immune proteins, suggesting that SARS-CoV-2 triggers well-defined immune responses already at the entry site. The identification of cytokines/chemokines and the pathophysiological features of the COVID-19-induced cytokine storm is relevant in order to predict clinical worsening, any requirement for hospitalization, intubation, or mortality [29]; after all, this feature has already been associated with the clinical outcome in many other viral infections, including HIV-1 [48] and monkeypox [49].

The assessment of cytokine secretion in saliva could, therefore, provide a further diagnostic tool to properly classify disease severity and progression.

Its suitability was validated from gene expression analyses in PBMCs of SD and MD patients, indicating the activation of innate and adaptive immune responses that were proportional to disease severity. Indeed, increased expressions of activation markers (CD38 and CD69) and interferon-stimulated genes (IFITM1 and IFITM3) were detected in SD patients.

Apart from configuring saliva cytokine and NA profiling as a suitable diagnostic tool for COVID-19, the present study identifies circulating miRNA saliva and plasma profiles associated with COVID-19 severity. Notably, such profiles do not represent a mere marker of the infection, as several differences were observed by comparing SD and MD patients. These results suggest that monitoring salivary miRNA expression could help foresee the outcome of COVID-19. However, the genes, miRNAs, and cytokines/chemokines herein analyzed are known to correlate to many diseases beyond COVID-19. Thus, the risk of identifying biological predictors not faithfully correlated to COVID-19 represents a potential limitation of the present study.

## 4. Material and Methods

### 4.1. Participants and Sample Collection

This study included a total of 20 COVID-19-positive patients (as determined by the SARS-CoV-2 molecular test of nasopharyngeal swabs) enrolled from a larger cohort hospitalized at the Infectious Diseases Unit, Policlinic “Riuniti” of Foggia (Italy) between 1 March and 31 May 2020. Patients were divided into mild patients (MD; *n* = 10) and severe patients (SD; *n* = 10) according to COVID-19 disease severity and following the National Institute of Health (NIH) guidelines for COVID-19 treatment. Age- and sex-matched uninfected subjects (HC) (*n* = 10) were included. HC were negative for SARS-CoV-2 molecular and anti-N ELISA tests. Blood and saliva samples were collected at hospital admission on the same day of testing by spitting after repeated mouth-washing with water. Participants were asked not to eat, drink, or smoke for at least 30 min prior to saliva collection. Saliva was incubated at 56 °C for 10 min and centrifuged at 6000× *g* for 10 min. Supernatants were stored at −80 °C until use. Plasma was obtained by centrifugation of whole blood at 1200× *g* for 10 min and stored at −20 °C until use. PBMCs were isolated as previously described [50] and stored at −150 °C. None of the subjects were vaccinated for SARS-CoV-2 at the time of sampling.

The protocol was approved by the local Medical Ethical and Institutional Review Board of Policlinic “Riuniti” of Foggia (protocol number 49/C.E./2021). Full informed consent was obtained from all subjects involved in the study, in agreement with the Declaration of Helsinki principles.

### 4.2. SARS-CoV-2 Quantification

Viral RNA was quantified in the saliva of COVID-19 patients by using the Maxwell RSC Viral Total Nucleic Acid Purification Kit (Promega, Fitchburg, WI, USA) with the Maxwell^®^ RSC Instrument (Promega, Fitchburg, WI, USA), as previously described [51]. Briefly, single-step, real-time RT-qPCR (GoTaq^®^ 1-Step RT-qPCR, Promega, Fitchburg, WI, USA) and the 2019-nCoV CDC qPCR Probe Assay kit specifically designed to target two regions of the nucleocapsid gene of SARS-CoV-2 (N1 and N2) were used on a CFX96 instrument (Bio-Rad, Hercules, CA, USA). An absolute viral copy of SARS-CoV-2 N gene quantification was performed by generating a standard curve from the quantified 2019-nCoV_N-positive Plasmid Control (IDT, Coralville, IA, USA). A cycle threshold (Ct) value of <40 was considered positive, based on CDC guidelines.

### 4.3. SARS-CoV-2 Virus Neutralization Assay

A neutralizing activity assay was performed as previously described [52].

Briefly, 50 μL of plasma, starting from a 1:10 dilution, and 50 μL of whole saliva were diluted 1:2 by serial twofold series in 96-well plates. Fifty microliters of SARS-CoV-2 TCID_50_ was added to each well and incubated for 2 h at 37 °C at 5% CO_2_. After incubation, 100 μL of the solution containing plasma and/or saliva and virus were transferred to microplates seeded with 2 × 10^4^ VeroE6 cells and incubated for 72 h at 37 °C and 5% CO_2_. At the end of incubation, cells were fixed with 4% formaldehyde 37% *m/v* (Merck KGaA, Darmstadt, Germany), for 20 min and stained with 0.1% *m/v* crystal violet solution (Merck KGaA Darmstadt, Germany). A positive titer was equal to or greater than 1:10 or 1:2 for plasma and saliva samples, respectively. Every test included plasma (1:10 dilution) or saliva controls (undiluted), as well as cell (VeroE6 cells alone) and viral controls (threefold series dilution).

### 4.4. MiRNA Expression in Saliva and Plasma Specimens

Maxwell^®^ RSC miRNA from plasma or a serum kit (Promega, Fitchburg, WI, USA) was used to extract miRNA from saliva and plasma using the Maxwell^®^ RSC Instrument (Promega, Fitchburg, WI, USA).

One-hundred nanograms of mature miRNAs were reverse-transcribed into first-strand cDNA in a 20 μL final volume at 37 °C for 60 min using a miScript II RT Kit (Qiagen, Venlo, The Netherlands), in accordance with the manufacturer’s protocol. The expression level of 84 miRNAs with antiviral and/or immunological functions was evaluated using a miRNA PCR Array (MIHS-105Z) Qiagen, Venlo, The Netherlands). Arrays were performed on the CFX ConnectTM Real-Time PCR system (BIO RAD, Hercules, CA, USA). Undetermined raw CT values were set to 35. The expression profile was analyzed using the PCR Array Gene Expression Analysis Software (Qiagen, Venlo, The Netherlands). For each miRNA, Ct values were transformed into relative quantities using a normalization factor RNU6-2 for saliva and plasma. A fold regulation of ±2.5 was considered positive.

### 4.5. Quantigene Plex Gene Expression Assay

Total RNA was extracted from PBMCs, as previously described [53], and 100 ng of RNA was used for gene expression analyses by QuantiGene Plex assay (Thermo Scientific, Waltham, MA, USA). This approach provided a fast and high-throughput solution for multiplex gene expression quantitation, allowing for the simultaneous measurement of 40 custom-selected genes of interest in a single well of a 96-well plate. The QuantiGene Plex assay is hybridization-based and incorporates branched DNA technology, which uses signal amplification for the direct measurement of RNA transcripts. Results were calculated relative to GAPDH, β-Actin, and PPIB as housekeeping genes and expressed as ΔCt.

### 4.6. Cytokine and Chemokine Measurements

A concentration of 27 cytokines/chemokines was assessed in the saliva and plasma using immunoassays formatted on magnetic beads (Bio-Rad, Hercules, CA, USA), according to the manufacturer’s protocol, via Luminex 100 technology (Luminex, Austin, TX, USA). Some of the targets that resulted were over-range, and an arbitrary value of 4000 pg/mL was assigned, while 0 pg/mL was attributed to values below the limit of detection.

### 4.7. Statistical Analyses

The student’s *t*-test was applied when appropriate to compare continuous and categorical variables. One-way ANOVA was applied for the parametric multiple comparison. A *p*-value < 0.05 was chosen as the cut-off for significance. Data were analyzed using GRAPHPAD PRISM version 11 (GraphPad software, La Jolla, CA, USA).

## 5. Conclusions

SARS-CoV-2 infection modulates the immune/miRNA balance at both the systemic and oral levels with an intensity that is proportional to disease severity. Monitoring these parameters over time can help predict disease outcome and identify new markers of disease progression. Saliva presents many advantages over blood, with its collection being easy, safe, non-invasive, and cost-effective. The assessment of salivary NA, cytokines, and miRNAs as biomarkers to monitor SARS-CoV-2 infection can represent a significant step forward in the diagnosis and prognosis of COVID-19. Further studies will be needed to validate these results and to verify if and how, following vaccine and/or therapy administration, such profiles are altered in SARS-CoV-2-infected and vaccinated subjects. Overall, these results advise about the possibility to employ saliva-based biomarkers, including microRNAs, to diagnose infections and outcomes using airborne methods, even in combination with other non-invasive bio-specimens, such as breath.

## Figures and Tables

**Figure 1 ijms-24-10992-f001:**
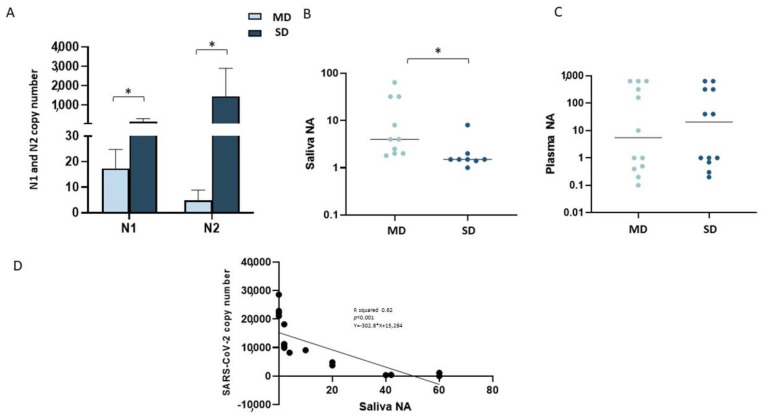
SARS-CoV-2 viral replication and neutralizing activity in saliva and plasma samples of MD and SD patients. Viral replications N1 and N2 of SARS-CoV-2 are reported in panel (**A**). Neutralizing activity (NA) from saliva and plasma of 10 MD patients (light blue bars) and 10 SD patients (blue bars), measured by virus neutralization assay (vNTA), are reported in panels (**B**) and (**C**), respectively. The Spearman correlation between the SARS-CoV-2 viral copy and NA level (**D**). Significance was indicated as follows: * = *p* < 0.05.

**Figure 2 ijms-24-10992-f002:**
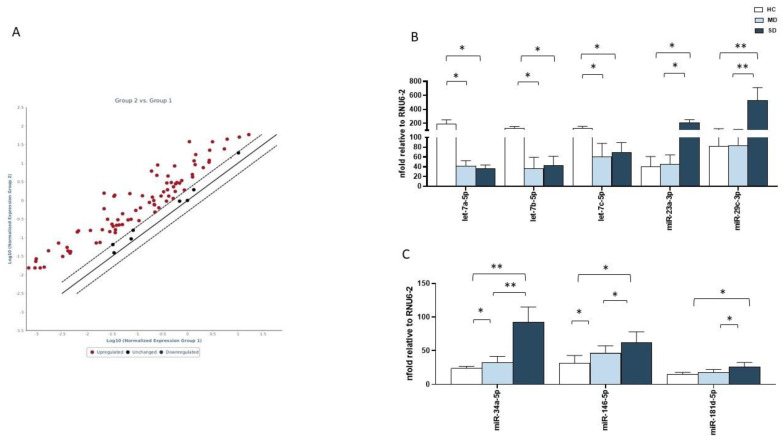
MiRNA expression analysis in saliva samples by PCR array. miRNA profile of SD patients, compared to MD patients (**A**). Analyses of miRNA with antiviral (**B**) and immunological functions (**C**) were performed in saliva samples of 10 HC (white bar), 10 MD patients (light blue bars), and 10 SD patients (blue bars). Results were calculated relative to the arithmetical mean of the references available in the arrays RNU6-2. Values were mean ± SEM. Significance was indicated as follows: * = *p* < 0.05, and ** = *p* < 0.01.

**Figure 3 ijms-24-10992-f003:**
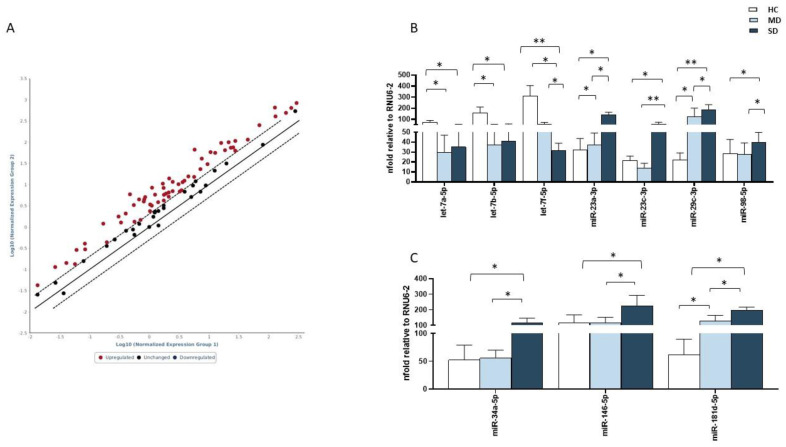
miRNA expression analysis in plasma samples by PCR array. miRNA profile of SD patients, compared to MD patients (**A**). Analyses of miRNA with antiviral (**B**) and immunological functions (**C**) were performed on saliva samples of 10 HC (white bar), 10 MD patients (light blue bars), and 10 SD patients (blue bars). Results were calculated relative to the arithmetical mean of the references available in the arrays RNU6-2. Values were mean ± SEM. Significance was indicated as follows: * = *p* < 0.05, and ** = *p* < 0.01.

**Figure 4 ijms-24-10992-f004:**
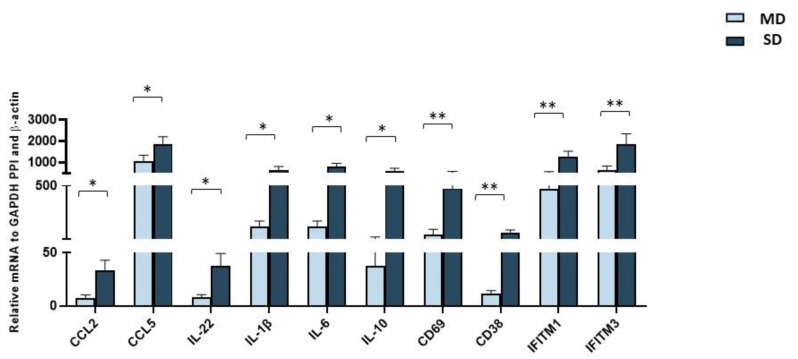
mRNA expression of genes involved in the antiviral/immune response. Quantigene Plex Gene expression technology was applied to quantify gene expression on RNA extracted from PBMCs isolated from 10 MD patients (light blue bars) and 10 SD patients (blue bars). Results were calculated relative to the arithmetical mean of the references available in the panel: GAPDH, b-Actin, and PPI. Only statistically significant *p*-values from the *t*-test comparison between SD and MD patients are shown * = *p* < 0.05, and ** = *p* < 0.01.

**Figure 5 ijms-24-10992-f005:**
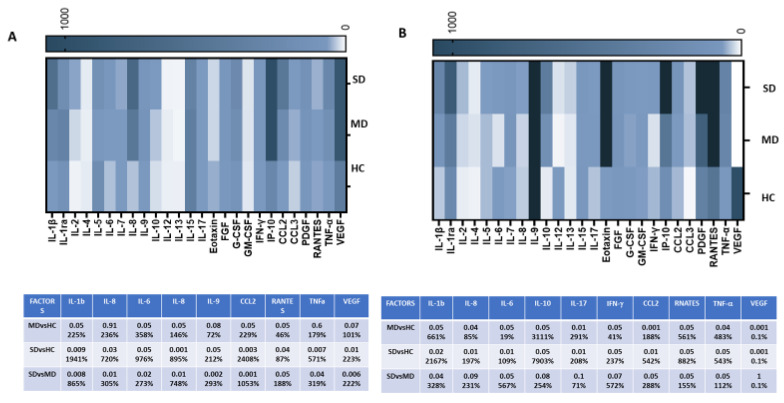
Plasma secretion of cytokines/chemokines that were part of the inflammatory response. The production of 27 cytokines/chemokines regulating immune response was assessed by Luminex assay in saliva (**A**) and plasma (**B**) specimens of 10 HC, 10 MD patients, and 10 SD patients. Cytokine/chemokine productions (mean values) are shown as a color scale from white to blue (heatmap). Only statistically significant p-values between at least 2 groups were reported, along with the percentage of difference in lower panels of (**A**, saliva) and (**B**, plasma).

**Table 1 ijms-24-10992-t001:** Enrolled subjects divided according to phenotype.

	Patients	Age	Sex	Therapy	Co-Infection
MILD		Mean ± SD 66.4 ± 16.97	F = 40%		
	1	45	F	1, 4	-
	2	57	F	1, 3	-
	3	85	F	0	-
	4	71	M	1, 2	-
	5	86	M	1, 2, 6	-
	6	59	F	1, 2	-
	7	70	M	1, 4	-
	8	40	M	1, 4, 5	-
	9	62	M	0	-
	10	89	M	1	-
**SEVERE**		**mean ± SD** **65.8 ± 12.20**	**F = 20%**		
	1	89	M	1	-
	2	49	M	1, 2, 3	-
	3	59	M	1, 4, 5	-
	4	49	F	1, 4, 5	-
	5	63	M	1, 2, 3, 4, 5	-
	6	75	M	1	-
	7	75	M	1, 2	-
	8	66	M	0	-
	9	64	F	1, 3	-
	10	69	M	0	-
**HC**		**mean ± SD** **62.5 ± 17.04**	**F = 60%**		
	1	52	F	0	-
	2	87	F	0	-
	3	74	M	0	-
	4	40	F	0	-
	5	55	M	0	-
	6	79	M	0	-
	7	44	F	0	-
	8	81	F	0	-
	9	47	F	0	-
	10	66	M	0	-

Therapy legend: 0 = No therapy; 1 = Hydroxychloroquin; 2 = Lopinavir; 3 = Darunavir-c; 4 = Darunavir/r; 5 = Tocilizumab; 6 = Other.

## Data Availability

The data that support the findings of this study are available from the corresponding author upon reasonable request. The data that support the findings of this study are available from the corresponding author upon reasonable request. The miRNA row data are available on NCBI platform with GSE accession number: GSE236017.

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
