# Peer review of "Salivary miRNA Profiles in COVID-19 Patients with Different Disease Severities"

_ijms, 2023, doi:10.3390/ijms241310992_

Round 1

Reviewer 1 Report

Comments and Suggestions for Authors

Thank you for the opportunity to review your manuscript. The study discussed in this manuscript explores how the oral mucosa plays a role in the early defense against SARS-CoV-2 infection. The researchers analyzed the viral load, miRNAs, cytokines, and neutralizing activity (NA) in the saliva and plasma of mild (MD) and severe (SD) COVID-19 patients. The manuscript is generally well written, scientifically sound, and provides valuable information to the reader. However, there are areas where the manuscript could be improved prior to publication. Here are my suggestions:

Major comments:

1.       It would be helpful to have a discussion on the differences in neutralizing antibody titers in saliva and blood between severe and mild cases, and the reasons for these differences.

2.       It is important to include comparative data or discussion on other viral and bacterial infections for all data presented in the manuscript.

3.       There is some confusion regarding whether the miRNA expression data is specific to SARS-CoV-2 or not, so addressing this concern would greatly improve the manuscript.

Minor comments:

1.       In line 97, please change the subscript of the '2' in CO2 to a subscript instead of a superscript.

Author Response

Thank you for the opportunity to review your manuscript. The study discussed in this manuscript explores how the oral mucosa plays a role in the early defense against SARS-CoV-2 infection. The researchers analyzed the viral load, miRNAs, cytokines, and neutralizing activity (NA) in the saliva and plasma of mild (MD) and severe (SD) COVID-19 patients. The manuscript is generally well written, scientifically sound, and provides valuable information to the reader. However, there are areas where the manuscript could be improved prior to publication. Here are my suggestions:

Major comments:

  1. It would be helpful to have a discussion on the differences in neutralizing antibody titers in saliva and blood between severe and mild cases, and the reasons for these differences.

We thank the Reviewer for the careful evaluation of the manuscript and the helpful suggestions to improve its quality. We tried our best to address, whenever possible, all the suggestions of the Reviewer. All the changes are marked in the revised version of the manuscript.

We thank the reviewer for her/his insightful comment. We also discussed this apparent inconsistency to find a plausible explanation as reported in the result section of the original manuscript (lines 187-196). However, we agreed to move our comment in the discussion session of the new version of the manuscript LINES 392-411

Intriguingly, the NA trend observed in the saliva of MD vs SD was opposite to that detected in the plasma. This inverse tendency has already been documented in other viral infections. For instance, Nekoua et al. reported that in coxsackievirus infection NA was significantly higher in saliva from diabetic patients compared to controls, but such differences were no longer evident at plasma level [50].

 It is tempting to speculate that the levels of NA in the plasma mirror the strong and generalized immune reaction (cytokine storm) that occurs at a systemic level following viral infection and tissue dissemination. Thus, it is conceivable that plasma NA levels are proportional to viral titers and to the severity of disease, being higher in SD compared to MD. Instead, NA in the saliva likely mirrors the pattern of an early, local immune response focused at controlling viral spread and tissue damage. In this case, the levels of NA might be inversely proportional to the severity of disease, being higher in MD compared to SD. However, the lack of statistical significance in salivary NA between MD and SD hinders any further speculations.

Accordingly, a paper from Valimaa et al. shows that in Herpes simplex virs (HSV) infection, NA in saliva from asymptomatic HSV-seropositive subjects is significantly higher compared to saliva NA in patients with recurrent labial herpes suggesting that salivary defense factors may contribute to control recurrence and presumably severity of infections [51].

  1. It is important to include comparative data or discussion on other viral and bacterial infections for all data presented in the manuscript.

The suggestion of the reviewer is appropriate and therefore we provided some comments all along the discussion section to include comparative data relative to other pathogens ( lines 392-397; 407-411; 418-421)

  1. There is some confusion regarding whether the miRNA expression data is specific to SARS-CoV-2 or not, so addressing this concern would greatly improve the manuscript.

We apologize for the confusion. The miRNA data were not SARS-CoV-2 specific or, at least, our data cannot prove a direct correlation between SARS-CoV-2 infection and miRNA production. Most of these miRNAs display antiviral or immunological functions. Indeed, their expression has already been widely documented in other pathological conditions including infectious ones, as reported in the discussion session. We, therefore, would speculate that this miRNA outline is triggered by the host response to any invading pathogen rather than a specific one. However, as their modulation was significantly different in COVID-19 patients compared to HC and in patients with different disease severity, it would be very useful to identify a miRNA combination suggestive of SARS-CoV-2 infection and outcome, especially in the saliva, to be used as predictive prognosis and diagnosis tool.

These observations have now been reported in the discussion section (Lines 377-386)

Minor comments:

  1. In line 97, please change the subscript of the '2' in CO2 to a subscript instead of a superscript.

We are sorry for the typos. The subscript has been changed to subscript

Reviewer 2 Report

Comments and Suggestions for Authors

The manuscript by Saulle et al studies the composition of miRNAs and cytokines found in saliva and plasma from patients infected with SARS-CoV2. they found that out of the 84 miRNA analyzed, 8 were present in patients with severe COVId-19, 3 downregulated and 5 upregulated. Also , the production of pro-inflammatory cytokines and chemokines were increased in both saliva and plasma of severe amd mild disease. The authors propose that monitoring these parameters coud help to predict disease outcome.

The introduction is sufficient, the methodology is well explained and the results are original and well presented including the statistics.

Results are interesting for scientist in the area, and the conclusions well supported.

My only major concern to improve the manuscript is that the results from the array (miRNA PCR Array (MIHS-106Z) (Quiagen) should be deposited in the GEO repository, get an GSE accesion number and include it in the manuscript. This will be informative for the scientific community, and can be done quickly. Authors can check in the GEO database observing that this specific platform is already cited there (Platform GPL32545).

Author Response

We are in complete agreement with Reviewer 2. We have submitted the data to GEO platform and now we are waiting for the GSE access number which will be included in the manuscript as soon as possible

Round 2

Reviewer 1 Report

Comments and Suggestions for Authors

Thank you for submitting the revised manuscript entitled " Salivary miRNA profiles in COVID-19 patients with different disease severity." I appreciate your effort in addressing the concerns and questions raised in my initial review. Your detailed explanations and clarifications have significantly improved the manuscript. I look forward to seeing it published.

Reviewer 2 Report

Comments and Suggestions for Authors

Ok, make sure to include the GSEnumber in the manuscript before publication